Autonomous intersection over union (IoU) loss: adaptive dynamic non-monotonic focal IoU loss

Zhu Yanchen 1
http://orcid.org/0009-0007-1002-809X Zheng Wenhua 1
Du Jianqiang 1 2
Huang Qiang 1 20201041@jxutcm.edu.cn
1 School of Computer Science, Jiangxi University of Chinese Medicine , Nanchang, Jiangxi , China
2 Nanchang University , Nanchang, Jiangxi , China
Angiulli Giovanni
Electronic publication date: 2024 Sep 26
Publication date: 2024
Volume: 10
Electronic Location ID: e2347
Received 2024 Mar 14; Accepted 2024 Aug 30
Copyright: © 2024 Zhu et al.
Copyright year: 2024
Copyright holder: Zhu et al.
License: This is an open access article distributed under the terms of the Creative Commons Attribution License, which permits unrestricted use, distribution, reproduction and adaptation in any medium and for any purpose provided that it is properly attributed. For attribution, the original author(s), title, publication source (PeerJ Computer Science) and either DOI or URL of the article must be cited.
License URL: https://creativecommons.org/licenses/by/4.0/

Keywords: Object detection, IoU, Loss function design, Focusing mechanism, Hard mining

Funding: National Natural Science Foundation of China 82260988, 82274680, and 82160955 National Key Research and Development Program of China 2019YFC1712300 Jiangxi Provincial Department of Education GJJ2200966 Jiangxi University of Chinese Medicine JZYC23S14 Jiangxi University of Chinese Medicine Science and Technology CXTD22015 Jiangxi University of Traditional Chinese Medicine 2023jzzdxk026 This work is supported by National Natural Science Foundation of China (Nos. 82260988, 82274680, and 82160955), the National Key Research and Development Program of China (No. 2019YFC1712300), the Jiangxi Provincial Department of Education (No. GJJ2200966), Jiangxi University of Chinese Medicine Graduate Innovation Special Foundation (No. JZYC23S14), the Jiangxi University of Chinese Medicine Science and Technology Innovation Team Development Program (No. CXTD22015) and the Project supported by funding for the construction of key disciplines at Jiangxi University of Traditional Chinese Medicine (No. 2023jzzdxk026). The funders had no role in study design, data collection and analysis, decision to publish, or preparation of the manuscript.

==============================
In object detection algorithms, the bounding box regression (BBR) loss directly influences the model’s accuracy in predicting object positions. Initially, we conducted simulation experiments on the proposed intersection over union-based loss. Through in-depth analysis, we discovered that the width and height components of the distance term could hinder BBR. To address this, we proposed the AIoU-v1 loss, which decouples the width and height components of the distance term, thereby preventing the suppression of BBR loss by the distance term. Additionally, issues such as the imbalance between sample quantity and sample quality in the dataset, as well as labeling errors, can adversely affect BBR. To tackle these dataset problems, we designed an adaptive dynamic non-monotonic focusing mechanism with strong robustness and wide applicability. Finally, we proposed a post-processing algorithm that combines fusion and non-maximum suppression, resulting in more accurate bounding boxes during the post-processing stage. Our source code and data are available at https://github.com/Wenhua-Zheng/AIOU.

Introduction

Object detectors are composed of three main components: the Backbone, Neck and Head (Bochkovskiy, Wang & Liao, 2020). The Head typically includes a classification branch and a localization branch. The loss function of a target detection model generally comprises classification loss, bounding box regression (BBR) loss, and target confidence loss (Redmon & Farhadi, 2018). The convergence performance of the BBR loss directly affects the model’s localization performance, which is a growing but challenging area of research (Yao et al., 2022). Therefore, improving the BBR loss is crucial for enhancing the model’s localization performance and ensuring effective convergence. An additional challenge in the post-processing stage is determining how to select the final bounding boxes from the numerous predicted boxes to improve the model’s localization accuracy.

Loss functions for BBR

The BBR losses in proposal, anchor, and anchor-free based models are mainly classified into two categories: Ln−norm and IoU losses. Early target detection models primarily use Ln−norm loss. Ln−norm loss computes the Euclidean distance and geometric difference between predicted boxes and ground truth (GT) boxes, and is commonly employed in early target detection models. The BBR loss can be simply described as:

(1) L(B,Bgt)=f(B,Bgt).

Most of the BBR loss value is derived from large targets and lacks sensitivity to small target losses, resulting in poor localization ability for small targets (Yu et al., 2016). Subsequent improvements involved using anchors in the calculation of BBR loss (Liu et al., 2016), which reduced the emphasis on large targets but depended heavily on the quality of the anchor settings. Furthermore, the positioning performance for small target outliers was not effectively enhanced. Early target detection models primarily used Ln−norm loss. R-CNN (Girshick et al., 2014) and SPP-Net (He et al., 2015) employed L2−norm loss, while subsequent models like Fast R-CNN (Girshick, 2015) and Faster R-CNN (Ren et al., 2015) used smoothL1 loss (Eq. (2)). The performance of smoothL1 loss is directly affected by the quality of the set anchor box, and the loss value remains small for small outlying targets, resulting in the regression effect of small targets is not good enough.

(2) smoothL1(t−tgt)={0.5(t−tgt)2,|x|<1|t−tgt|−0.5,otherwise.

To address the aforementioned problem, Yu et al. (2016) proposed Intersection over Union (IoU) loss (Eq. (3)). IoU is used to compute the overlap between predicted boxes and GT boxes, and IoU losses have been incorporated into many subsequent target detection models (Yu et al., 2016), such as the YOLO series and FCOS (Tian et al., 2019). The loss function is defined as:

(3) LIoU=1−IoU=1−B∩BgtB∪Bgt.

IoU Loss focuses on the ratio between the intersection and union of predicted boxes and GT boxes, thereby avoiding the poor localization ability of Ln−norm loss for small targets. However, when B∩Bgt=0, the gradient back-propagated by LIoU vanishes (Rezatofighi et al., 2019). To address this issue, many improvements to IoU-based loss functions have been proposed, such as GIoU Loss (Rezatofighi et al., 2019) and DIoU Loss (Zheng et al., 2020). These improved loss functions (Eq. (4)) add a penalty term R(B,Bgt) to the IoU loss to enhance localization capability.

(4) L(B,Bgt)=1−IoU+R(B,Bgt).

Generalized IoU loss (GIoU loss), proposed by Rezatofighi et al. (2019), effectively addresses the drawbacks of IoU. Zheng et al. (2020) proposed Distance-IoU loss (DIoU loss), which uses the central regression term (distance term) as the penalty term for IoU loss. This resolves the issue where the gradient of GIoU loss is very small when IoU = 0 and the difference between the union and intersection sets is close to zero. Complete-IoU loss (CIoU loss) (Zheng et al., 2020) incorporates the ratio of borders between predicted boxes and GT boxes as a second penalty term based on DIoU loss. Zhang et al. (2022) further modified the penalty term for the border ratio in CIoU loss, and Efficient IoU loss (EIoU loss) compares the geometric difference between two borders more intuitively. Additionally, SIoU loss (Gevorgyan, 2022), alpha-IoU loss (He et al., 2021), and WIoU loss (Tong et al., 2023) have been proposed to further enhance IoU loss. In the BBR process, the gradients of x and y in the central regression term are directly related to the width ( w) and height ( h) of the bounding box. When w is much larger than h, the denominator value is primarily dominated by w, which severely suppresses the gradient of y, and vice versa. We hypothesize that decoupling the distance terms of DIoU, EIoU, and WIoU losses is more advantageous for bounding box regression. This hypothesis has been experimentally validated in Figs. 1 and 2. Recent studies related to BBR loss (Wen et al., 2022; Cheng et al., 2022; Yuan et al., 2021) have focused on improving the localization performance of the model.

Figure 1 Space simulation experiment.

Figure 2 Point cloud maps of regression error were obtained by visualizing the total final error, which is the sum of regression errors for all cases in the simulation space, after 200 epochs of regression on 5,000 discrete points.

Focusing mechanism

There is an imbalance between the number and quality of samples in the dataset for target detection (Northcutt, Jiang & Chuang, 2021). Focusing too much on high-quality samples exacerbates the imbalance in sample numbers, and outliers among low-quality samples generate numerous harmful gradients, negatively affecting the model’s convergence performance (Pang et al., 2019).

During training, more focus should be placed on samples of ordinary quality, allocating larger gradients to them. Conversely, the gradients for both high-quality and low-quality samples should be reduced. The existing focusing mechanism (FM), which concentrates on loss, aims to improve regression by focusing on certain anchor boxes to address issues like effective example mining. It achieves gradient allocation by reweighting the loss, but the weighting curve is fixed and cannot be adjusted according to the training period and convergence results.

In addition to issues of sample quantity and quality imbalance, problems such as missed labeling and false positive labeling in the dataset can seriously jeopardize Bounding Box Regression (BBR) performance. Existing focusing mechanisms can only alleviate these labeling issues for non-dense targets. Addressing the imbalanced dataset and annotation problems mentioned above is key to improving BBR performance. Therefore, we proposed a new focusing mechanism (Eq. (25)).

In the early stages of training, the focusing mechanism should improve overall sample quality by setting a very low threshold for low-quality samples, gradually increasing this threshold as training progresses. Similarly, in the later stages of training, we should raise the threshold for high-quality samples because, by then, the overall sample quality has greatly improved. Focusing more on improving medium to high-quality samples at this stage can lead to better mean average precision (mAP) for the model.

Finally, relying solely on IoU for the focusing mechanism cannot alleviate the problem of annotation offset in the dataset, nor can it address the issues of missing and mislabeling in dense targets. These issues can increase harmful gradients and ultimately harm BBR performance. Therefore, we proposed a more robust focusing mechanism (Eq. (27)). Our proposed method significantly improves the mAP values of the model across various quality categories, especially for medium quality categories.

Non-maximum suppression

During forward propagation, the model predicts multiple bounding boxes, which necessitates the use of non-maximum suppression (NMS) (Neubeck & Van Gool, 2006) to remove redundant and erroneous predictions and obtain the final bounding boxes. In the classic NMS algorithm, IoU is the only reference factor. The algorithm sorts the predicted boxes from high to low based on their confidence levels and iteratively calculates the IoU between the maximum confidence predicted boxes (MC-boxes) and other non-maximum confidence predicted boxes (NMC-boxes). When the IoU exceeds a certain threshold, the NMC-boxes are considered redundant and are eliminated. However, in practical applications, particularly with dense targets, this approach can lead to missed detections. Therefore, to improve the accuracy of NMS, it is necessary to introduce additional reference factors.

We believe that if the IoU value between MC-boxes and NMC-boxes is near the critical threshold and their confidence scores are similar, then the NMC-box is highly likely to represent another target. Therefore, the suppression score of this NMC-box should be reduced. Conversely, if the suppression score of the NMC-box is high, it is likely that this NMC-box and the MC-box represent the same target, and the position and boundary of this NMC-box have a certain reference value for the MC-box. We proposed the inhibition score AIoUFNMS (Eq. (30)) to achieve these functions and improve the performance of NMS.

We proposed an IoU-based loss and Autonomous FM, using Autonomous FM to weight the IoU-based loss, termed as Autonomous-IoU loss (AIoU loss), and improve the NMS algorithm during post-processing. We use simulation experiments and the detector YOLOv7 (Wang, Bochkovskiy & Liao, 2023) to test the actual performance of AIoU loss. The main contributions of this article are as follows: a) We deeply analyzed the original BBR loss and proposed AIoU-v1 loss, which has a faster convergence speed in simulation experiments and higher Mean Average Precision (mAP) in model training.

b) We designed AIoU-v3 with different adaptive dynamic non-monotonic FM for different categories, allowing the model to have more reasonable gradient allocation during the training process and further addressing the imbalance between the number of samples and the quality of the samples.

c) We compared the effects of introducing IoU and confidence on BBR in FM, and then designed AIoU-v4 by introducing confidence in AIoU-v3, which further improves the model’s ability to withstand false positive labeling information in the dataset and reduces the detrimental effects of harmful gradients on the model’s BBR.

d) We modified the original NMS algorithm and proposed AIoUFNMS to enable the model to further improve localization capability during post-processing.

Related work

In this section, we first analyze the characteristics of existing popular loss functions, focusing mechanisms, and Non-Maximum Suppression.

Limitations on various IoU losses

Limitations of IoU and GIoU losses

Yu et al. (2016) proposed IoU loss (Eq. (3)). The analysis of Eq. (3) shows that when IoU=0, the IoU loss suffers from the problem of vanishing gradients, making it impossible to optimize the regression parameters of the bounding box. This issue is illustrated in Figs. 2A, 3B and 3D. To address the problem of vanishing gradients in IoU loss, Rezatofighi et al. (2019) proposed GIoU loss:

(5) LGIoU=1−IoU+|C−B⋃Bgt||C|

Figure 3 In the simulation experiments with bounding box regression, the left side of the graph shows the anchor box regression error, while the right side depicts the regression performance of the anchor box at the 150th epoch.

C is the smallest enclosing convex object of B and Bgt. GIoU loss compensates for the vanishing gradient issue of IoU loss when IoU=0, as shown in Fig. 2B.

When |C−B⋃Bgt| is smaller, the GIoU loss is closer to the IoU loss, and when |C−B⋃Bgt| is close to 0, the GIoU loss nearly degrades to the IoU loss. From Fig. 2B, it can be seen that when the included angle is 45∘ (the included angle between the center point of GT boxes and anchor boxes and the axis is 45∘), there are four valley regions, and regression is particularly fast at this time. The smaller the included angle, the slower the regression of the bounding box, making it difficult to optimize the regression parameters when the included angle is 0∘. This analysis is also supported in Fig. 3, where the regression performance of GIoU is normal when the included angle is 45∘ (Figs. 3B and 3D), but poor when the included angle is 0∘ (Figs. 3A and 3C).

Limitations of distance IoU loss

Zheng et al. (2020) introduced the central regression terms of GT boxes and anchor boxes based on IoU loss and proposed the DIoU loss:

(6) LDIoU=1−IoU+ρ2(x−xgt)+ρ2(y−ygt)wc2+hc2.

The introduction of this penalty term solves the problem of the vanishing gradient of IoU loss and the tiny gradient of GIoU loss. However, as mentioned in WIoU loss, wc and hc can hinder the regression of GT boxes and anchor boxes. Therefore, we set the gradient of wc and hc of the centroid distances to zero during the bounding box regression process. In Fig. 1B, it can be seen that the DIoU loss is more stable and faster than the GIoU loss, without serious gradient decay. In Figs. 2C and 3, DIoU loss can complete the bounding box regression at any included angle and solve the hidden problems of IoU and GIoU losses.

Limitations of complete IoU loss

Zheng et al. (2020) proposed CIoU loss by adding the aspect ratios of GT boxes and anchor boxes to DIoU loss:

(7) LDIoU=1−IoU+ρ2(x−xgt)+ρ2(y−ygt)wc2+hc2+αV

(8) V=4π2(arctan⁡wgthgt−arctan⁡wh)2

(9) α={0,IoU<0.5V(1−IoU)+V,IoU≥0.5.

In EIoU loss, the authors point out that the V term in the last loss term of CIoU loss is not well defined, leading to the convergence performance of CIoU loss not being significantly improved compared to DIoU loss. In Fig. 2D, it can be seen that the convergence effect of CIoU loss is not significantly better than that of DIoU loss (Fig. 2C), and it also cannot address the slower convergence of DIoU loss when the included angle is 45∘.

Limitations of efficient IOU loss

Zhang et al. (2022) modified the aspect term of CIoU loss and proposed EIoU loss:

(10) LEIoU=LIoU+Ldis+Lasp=1−IoU+ρ2(x−xgt)+ρ2(y−ygt)wc2+hc2+ρ2(w−wgt)wc2+ρ2(h−hgt)hc2.

The aspect term of EIoU loss represents the difference in height and width between GT boxes and anchor boxes more directly, leading to faster and more accurate convergence of anchor boxes. After introducing the aspect term, it can be seen in Fig. 1B that EIoU loss has a faster convergence rate compared to previous BBR losses. In Figs. 2E and 3, it can be observed that the cumulative regression losses of GT boxes and anchor boxes are smaller than those of IoU, DIoU, and CIoU losses at any relative position.

Limitations of wise-IOU loss

Tong et al. (2023) proposed wise IoU (WIoU) loss (Eq. (11)), which modifies the distance term of DIoU loss by changing the distance term from a bias to a weight to weight the IoU loss. Additionally, it uses an exponential function to increase the gradient during regression.

(11) LWIoU−v1=RWIoULIoU

(12) RWIoU=exp⁡(ρ2(x−xgt)+ρ2(y−ygt)wc2+hc2).

Compared to DIoU loss, the distance term of WIoU loss not only aids in the regression of the center position but also allows the distance term and the IoU term to weigh each other. When the gradient of one term is larger, the gradient of the other term is also larger. In Figs. 1B and 3, it can be seen that the regression speed of WIoU loss is better than that of all previous BBR losses. In Fig. 2F, the regression loss of WIoU loss has a lower value in the loss of point cloud maps as well. However, the reason for the lower mAP of WIoU-v1 loss in actual model training remains unknown.

Limitations on various FMs

Northcutt, Jiang & Chuang (2021) noted the inherent imbalance between the number of samples and their quality, as well as issues like missed labeling and false positive labeling in target detection datasets. During training, categories with larger sample sizes tend to receive higher quality, while categories with fewer samples receive lower quality. Suppressing the loss of high-quality samples can help mitigate the issue of imbalanced sample sizes, while suppressing the loss of low-quality samples can address problems related to missing and mislabeled data.

In Focal-EIoU-v1 (Eq. (10)) loss, which employs the non-monotonic focusing mechanism (hereafter referred to as E-FM), the IoU value is used to weight the EIoU loss, and E-FM effectively suppresses all sample losses.

(13) LFocal−EIOU=IoUγLEIoU.

The Focal-EIoU-v1 Loss does not effectively determine the threshold for low-quality samples. In contrast, the WIoU-v2 loss (Eq. (14)) adjusts the weighting term based on E-FM by using the current overall mean of 1-IoU as a baseline. This approach helps in establishing the threshold for identifying high-quality samples.

(14) LWIoU−v2=(LIoU⋆LIoU¯)γLWIoU−v1,γ=0.5.

This modification fully leverages the potential of monotonic FM (i.e., WIoU-v2 loss of FM, hereafter referred to as W2-FM), which dynamically calculates thresholds for high-quality and low-quality samples, accurately suppressing the loss of high-quality samples.

WIoU-v3 loss (Eq. (15)) improves the speed and accuracy of bounding box regression by introducing a new weighting curve (hereafter referred to as W3-FM). This curve suppresses the loss of both high-quality and low-quality samples while increasing the loss for medium-quality samples, addressing the limitation of W2-FM, which only suppresses the loss of high-quality samples.

(15) LWIoU−v3=γLWIoU−v1,γ=βδαβ−δ,β=LIoU⋆LIoU¯.

However, WIoU-v3 loss has the following shortcomings: a) The weighted curve is fixed and cannot be adjusted based on the training period and regression performance.

b) If the values of α and δ are not suitable, they may hinder the model’s convergence. For different datasets, determining more appropriate α and δ values relies on numerous repeated experiments, which means its generalization still needs optimization.

c) Fixed α and δ values are not applicable throughout the entire training process, as the overall quality of samples varies at different stages of training, affecting the thresholds for high- and low-quality samples.

d) The baseline for sample quality is unreasonable. The quality of each sample should be evaluated using the category of the current sample as a baseline rather than an overall assessment. This is due to differences in the quality of bounding box regression for each category. If the regression quality is poor for a category, using the overall bounding box regression quality as the baseline will severely suppress the regression gradient for that category, and vice versa.

Limitations on various NMS algorithms

In practical target prediction, except for anchor-free target detection algorithms (e.g., Faster R-CNN, SSD, YOLO), the same target typically predicts a very large number of bounding boxes. This necessitates the use of the Non-Maximum Suppression (NMS) algorithm to obtain the final boxes from all predicted ones.

The traditional NMS algorithm, which determines whether a predicted box is redundant based on the IoU (Intersection over Union) value, is not suitable for detecting overlapping and dense targets. Soft-NMS Bodla et al. (2017) addresses this issue by indirectly excluding redundant boxes, reducing the confidence of the predicted boxes through the IoU value. Softer-NMS He et al. (2018) models the probability distributions of the predicted boxes’ positions. Based on the degree of overlap and positional uncertainty, it applies a voting mechanism to the overlapping predicted boxes. Boxes with a high degree of overlap and small variance in location distribution are given more weight, resulting in more accurate predictions. Weighted Boxes Fusion (WBF) Solovyev, Wang & Gabruseva (2021) fuses all boxes to obtain final boxes, solving both the dense target detection problem and the issue of no direct correlation between confidence and localization accuracy.

Method

In this section, we first analyze the characteristics of existing popular loss functions and then proposed the Autonomous-IOU loss.

Simulation experiment

To compare the convergence ability of each loss function for BBR, we analyzed the loss space using the simulation experiment proposed by Zheng et al. (2020). We adhered to the original simulation environment, including bounding boxes with respect to distance, scale, and aspect ratio, as shown in Fig. 1A. Given a loss function, we simulated the bounding box regression process for each case using the stochastic gradient descent algorithm. We first performed 200 epochs of regression on six representative IoU losses (IoU, GIoU, DIoU, CIoU, EIoU, WIoU) and three IoU-de losses (DIoU-de, EIoU-de, WIoU-de, which are derived by decoupling the height and width from the distance term in the DIoU, EIoU, and WIoU loss functions, respectively). We separated the distance terms to obtain point cloud maps of the total final regression error (Fig. 2) and regression error curves (Fig. 1B).

In addition to conducting spatial loss simulation experiments, it is also necessary to study the regression behavior in specific orientation and overlap relationships between anchor boxes and target boxes. We conducted BBR simulation experiments for two different orientations (the line connecting the center points of the target box and the anchor box forms an angle of 45∘ or 0∘ with the axis), considering both intersecting and non-intersecting scenarios (whether the anchor box and the target box overlap or do not overlap). This setup is illustrated in Fig. 3.

The proposed method

Designing a more reasonable IoU-based loss

To validate our reasoning, we conducted the same simulation experiment by separating the distance terms of DIoU, EIoU, and WIoU losses to obtain DIoU-de, EIoU-de, and WIoU-de losses. In Fig. 2G, when the angle is around 45∘, the convergence speed and overall bounding box regression performance of the DIoU-de loss are greatly improved, even surpassing that of the CIoU loss. In Figs. 1B, 2H, 2I and 3, the regression performance of the EIoU-de and WIoU-de losses also showed improvements. We proposed the AIoU-v1 loss:

(16) LAIoU−v1=RAIoULIoU

(17) RAIoU=exp⁡(ρ2(x−xgt)wc2+ρ2(y−ygt)hc2).

Through simulation experiments, we compare the regressions of various IoU-based losses, and we obtain the following conclusions: a) Among the BBR losses discussed in existing work, modifying the WIoU-v1 loss to create our AIoU-v1 loss (i.e., WIoU-de loss) results in the fastest convergence rate, as shown in Fig. 3.

b) The convergence performance of all BBR losses is very similar, with differences primarily arising from non-overlapping bounding boxes. In the YOLO series, non-overlapping bounding boxes are rare because anchor boxes have predefined precincts, so this does not lead to significant performance improvement during actual training. However, our proposed AIoU-v1 loss is expected to bring greater performance improvements for SSD and DETR (Carion et al., 2020).

Optimizing unilateral suppression

In “Limitations of Wise-IoU Loss”, we analyzed that it is more reasonable to suppress low-quality samples by the weighting term, and proposed a dynamic non monotonic focusing coefficient (IoU⋆IoU¯)γ (hereafter referred to as A2-FM) and AIoU-v2 loss for suppressing low-quality samples:

(18) LAIoU−v2=(IoU⋆IoU¯)γLAIoU−v1,γ=0.5

IoU¯∈[0,1], this item is the mean value of the overall IoU during training. In our experiments, we found that dynamically updating the γ coefficients according to IoU¯ makes it possible to more accurately suppress low-quality samples during the training process and accelerate the convergence of the model, but its performance is not stable, and it is necessary to select a more appropriate functional relationship to better improve the performance of BBR.

Designing bilateral suppression

We proposed AIoU-v3-init loss (Eq. (19)) (the initial version of AIoU-v3 loss), which updates the weighting curves in real time based on the value of IoU¯, enabling the training process to autonomously differentiate between high quality, average quality and low quality samples.

(19) LAIoU−v3−init=γLAIoU−v1,γ=βδαβ−δ,β=LIoU⋆LIoU¯.

After rigorous theoretical analysis, we set IoU thresholds for high-quality samples and low-quality samples.

The IoU threshold for high-quality samples is:

(20) 0.5(IoU¯+1).

The IoU threshold for low-quality samples is:

(21) f(IoU¯)={0.076,IoU¯<0.41.5IoU¯3−0.5IoU¯2+0.1(1−IoU¯),otherwise.

The curvilinear relationship between the IoU threshold for low-quality samples and IoU¯ is shown in Fig. 4A.

Figure 4 Function curve.

As the model is optimized, the thresholds for high-quality and low-quality samples increase as well. When the IoU of a sample IoU∈[0.5(IoU¯+1),f(IoU¯)], the gradient of that sample is increased and vice versa.

The threshold for low-quality samples is:

(22) g(IoU¯)={1.54,IoU¯<0.41.5IoU¯2+IoU¯+0.9,otherwise.

When βδαβ−δ=1, the threshold for high-quality samples is the solution at the left endpoint (i.e., β1=0.5), and the threshold for low-quality samples is the solution at the right endpoint (i.e., β2=g(IoU¯)). δ determines the threshold for low-quality samples at the right endpoint. where δ makes γ=1 when β=δ, which yields: β=0.5, δ=1.5IoU¯2+IoU¯+0.9, α=(βδ)1β−δ. Thus, the values of both α and δ are only affected by IoU¯, and the weighting curves vary adaptively with IoU¯, as show in Fig. 4B.

In “Limitations of Wise-IoU Loss”, we analyzed that the weighting curve should increase the overall regression gradient in the pre-training phase to allow the model to have a wider range of learning abilities, and as the IoU¯ improves, the weighting curve gradually returns to normal gradient enhancement and suppression. Therefore, we proposed the AIoU-v3-opt loss (Eq. (23)) (an optimized version of AIoU-v3-init loss).

(23) LAIoU−v3−opt=γLAIoU−v1,γ=β∗μ(IoU¯+0.2)δαβ∗μ−δ

(24) μ=11−0.8∗(1−β).

In “Limitations of Various Fms”, we also analyzed that the learning effect of each category feature is directly related to the number and quality of samples in the dataset for that category, but we use the IoU¯ of all samples as a baseline to calculate the quality of each sample. This is actually unreasonable because for categories with low quantity and quality, their gradients will always be suppressed. Therefore, we propose AIoU-v3 loss (Eq. (25)), which independently calculates the IoUc¯ of each category to obtain independent weighted curves (hereafter referred to as A3-FM).

(25) LAIoU−v3=γcLAIoU−v1,γc=βc∗μc(IoUc¯+0.2)δαβc∗μ−δ

(26) μc=11−0.8∗(1−βc).

Independently calculating the weighted curves for each category, AIoU-v3 loss improved mAP by 0.09% compared to AIoU-v3 opt loss. The further optimized FM for AIoU-v3 loss not only no longer needs to repeat multiple experiments to select more appropriate α and δ coefficients for the dataset, but also has more reasonable gradient magnitudes throughout the training phase (In Fig. 4C). In the mid and late stages of training, AIoU-v3 loss will reduce the gradient of low-quality samples to minimize harmful gradients, while also focusing on medium-quality samples. On the MS-COCO2017 and VOC2012 datasets, AIoU-v3 loss improved the mAP of the model by 0.61% and 0.74%, respectively.

Introducing confidence into FMs

In the dataset, many targets have labeling offset, the IoU loss values of those anchor boxes and GT boxes may then be large, and harmful gradients will be generated. And for the problem of false positive labeling and missed labeling of dense targets, the focusing mechanism, such as WIoU-v3 and AIoU-v3 losses, which have been mentioned before, will only further increase its gradient.

The problem of mislabeling is also mentioned in Northcutt, Jiang & Chuang (2021), where it is mentioned that many people have proposed a solution called “confidence learning”. Similarly, we introduce confidence into the IoU loss computation. Similar to the strategy of AIoU-v3 loss, we suppress the loss values of samples with relatively too high and too low confidence, and pay more attention to samples whose confidence is around the mean. We proposed AIoU-v4 loss (the FM of AIoU-v4 loss hereafter referred to as A4-FM):

(27) LAIoU−v3=γcηcLAIoU−v1

(28) γc=βc,IoU∗μc,IoU(IoUc¯+0.3)δIoUαIoUβc,IoU∗μc,IoU−δIoU,ηc=βc,conf∗μc,conf(confc¯+0.3)δconfαconfβc,conf∗μc,conf−δconf

(29) βc,IoU=LIoU⋆Lc,IoU¯,βc,conf=Lconf⋆Lc,conf¯,μc,IoU=11−0.8∗(1−βc,IoU),μc,conf=11−0.8∗(1−βc,conf).

The likelihood that the prediction for that target is correct is determined by the value of the confidence level, and this likelihood is converted into a weighting factor to reweight the AIoU-v3 loss. In this way, the gradients generated for both low confidence targets (labeling offset targets and false positive labeling of dense targets) and high confidence targets (missed labeling of dense targets) are suppressed, which enables further resolution of the dataset labeling error problem. On the MS-COCO2017 and VOC2012 datasets, AIoU-v4 loss improved the mAP of the model by 0.79% and 1.01%, respectively. AIoU-v4 loss significantly improves the mAP value of the model for medium-quality categories.

Designing post-processing algorithms

We proposed Fusion and Non-Maximum Suppression (FNMS), which takes the IoU values between predicted boxes as the main basis of determination, and weighs the IoU by the ratio of confidence to suppress the suppression scores of non-maximum predicted boxes, and at the same time adjust the locations and bounding boxes of maximum predicted boxes by combining with WBF. We proposed the suppression score AIoUFNMS:

(30) AIoUFNMS=IoU⋅Rconf

(31) Rconf=0.2cos⁡π2θ,θ=confmaxconfnon−max.

It suppresses non-maximal predicted boxes if the score of AIoUFNMS is below a certain threshold, otherwise, it performs WBF on the maximal prediction frame, as shown in Eq. (32).

(32) {NMS,AIoUFNMS<thresholdWBF,otherwise

On the MS-COCO2017 and VOC2012 datasets, AIoUFNMS loss improved the mAP of the model by 0.86% and 1.12%, respectively.

Experiments

Experimental setup

In order to fairly compare the performance of each loss function, we adopted the same training and validation environments. We used two publicly available datasets. The first dataset consisted of 16,000 images selected from the MS-COCO dataset (Lin et al., 2014) as the training set and 3,000 images as the validation set, with a total of 80 categories. The second dataset is the complete VOC2012 dataset (Everingham et al., 2015) with 20 categories. YOLOv7 is trained for 60 epochs with different BBR losses.

Ablation study

In the simulation experiments, we found that separating the x and y components of the center regression terms in IoU losses can prevent interference between the x and y regressions, thereby speeding up the bounding box regression. We compared the regressions of IoU, DIoU, and WIoU losses, as well as the regressions of IoU losses with the distance terms separated (DIoU-de and WIoU-de losses), as shown in Table 1. The performance of DIoU and WIoU losses showed a slight improvement over the IoU loss, while DIoU-de and WIoU-de losses further enhanced performance based on DIoU and WIoU losses.

Table 1 Performance of each IoU loss on the MS-COCO (the decoupling distance terms of DIoU and WIoU are shown in bold, and their regression performance has been improved).

BBR loss	mAP	AP50	
IoU	42.29	61.02	
DIoU	42.49 (+0.20)	60.92 (−0.10)	
DIoU-de	42.58 (+0.29)	61.10 (+0.08)	
WIoU	42.37 (+0.08)	61.18 (+0.16)	
WIoU-de	42.44 (+0.15)	62.70 (+1.68)	

WIoU-v2 loss introduces a new Focus Metric (FM) that uses the current overall mean IoU as a reference to assess the quality of the current sample. We conducted an ablation experiment to compare the regression performance of weighting IoU, DIoU-de, and WIoU-de losses using the FMs of W2-FM and A2-FM, as shown in Table 2. The experimental results indicate that suppressing low-quality samples (AIoU-v2 FM) is more effective than suppressing high-quality samples (WIoU-v2 FM). Specifically, the AIoU-v2 loss enhances the performance of bounding box regression more than the WIoU-v2 loss. When applied to the IoU loss, its performance is improved by 0.53.

Table 2 Performance loss of each IoU under the W2-FM and A2-FM on the MS-COCO (the results of using A2-FM weighted IoU, DIoU, and WIoU are shown in bold, and their regression performance has been improved).

BBR loss	FMs	mAP	AP50	
IoU	None	42.29	61.02	
IoU	W2-FM	42.55 (+0.26)	61.07 (+0.05)	
IoU	A2-FM	42.85 (+0.53)	61.27 (+0.25)	
DIoU-de	W2-FM	42.22 (−0.07)	60.99 (−0.03)	
DIoU-de	A2-FM	42.60 (+0.31)	61.17 (+0.15)	
WIoU-de	W2-FM	42.29 (+0.00)	61.19 (+0.17)	
WIoU-de	A2-FM	42.69 (+0.40)	61.13 (+0.11)	

When designing the A2-FM, we considered dynamically updating the γ coefficient according to IoU¯ so that the A2-FM would provide a more reasonable gradient assignment. However, in practice, the performance was not stable, and the bounding box regression performance was highly dependent on the design of the γ coefficients. As shown in Table 3, when γ=IoU¯−0.4, the model’s mAP improved by 0.05% and the model’s AP50 increased by 0.18% compared to γ=0.5. However, the regression performance of the model decreased when γ=IoU¯−0.3 and γ=IoU¯−0.5.

Table 3 Performance of WIoU-de loss under different dynamic coefficient FMs on the MS-COCO (the best results are shown in bold).

BBR loss	FMs	γ	mAP	AP50	
WIoU-de	A2-FM	0.5	42.69	61.13	
WIoU-de	A2-FM	IoU¯−0.3	42.46 (−0.23)	61.08 (−0.05)	
WIoU-de	A2-FM	IoU¯−0.4	42.74 (+0.05)	61.31 (+0.18)	
WIoU-de	A2-FM	IoU¯−0.5	42.40 (−0.29)	60.91 (−0.22)	

In practice, we can change the baseline in the AIoU-v2 loss from IoU¯ to conf¯, resulting in the AIoU-v2-conf loss. In Table 4, the mAP values obtained from the regressions using AIoU-v2-conf loss and AIoU-v2-IoU loss are not significantly different. However, during the training process, we found that the AIoU-v2-conf loss further improves the overall IoU value, while the AIoU-v2-IoU loss enhances the overall confidence value, as illustrated in Fig. 5.

Table 4 Performance loss of each IoU loss under the A2-FM of mean IoU and mean conf on the MS-COCO (the best results are shown in bold).

BBR loss	FMs	Baseline	mAP	AP50	
IoU	None	None	42.29	61.02	
IoU	A2-FM	IoU¯	42.85 (+0.53)	61.27 (+0.25)	
IoU	A2-FM	conf¯	42.61 (+0.32)	60.90 (−0.12)	
DIoU-de	A2-FM	IoU¯	42.60 (+0.31)	61.17 (+0.15)	
DIoU-de	A2-FM	conf¯	42.60 (+0.31)	60.97 (−0.05)	
WIoU-de	A2-FM	IoU¯	42.69 (+0.40)	61.13 (+0.11)	
WIoU-de	A2-FM	conf¯	42.62 (+0.33)	60.91 (−0.11)	

Figure 5 The impact of different FMs on model IoU and confidence during training.

We compared the performance improvement effects of the AIoU-v3 opt loss and AIoU-v3 loss on the model. The experimental results, shown in Table 5, indicate that separating the weighted curves for each category is more beneficial for bounding box regression.

Table 5 Performance of AIoU-v3-opt loss and AIoU-v3 loss on the VOC2012 (the best results are shown in bold).

BBR loss	mAP	AP50	
IoU	60.34	80.85	
AIoU-v3-opt	60.99 (+0.65)	80.87 (+0.02)	
AIoU-v3	61.08 (+0.74)	81.05 (+0.30)	

We compared the effectiveness of FM with WIoU-v3 loss ( α=1.9,δ=3.0) and AIoU-v3 loss in improving model performance. The experimental results, shown in Table 6, indicate that both W3-FM and A3-FM improve the model’s mAP values. Furthermore, A3-FM loss effectively addresses the limitations of fixed weighted curves and unreasonable sample quality benchmarks in W3-FM loss.

Table 6 Performance loss of each IoU loss under W3-FM and A3-FM on the MS-COCO (the best results are shown in bold).

BBR loss	FMs	Baseline	mAP	AP50	
IoU	None	None	42.29	61.02	
IoU	W3-FM	IoU¯	42.69 (+0.40)	61.35 (+0.33)	
IoU	A3-FM	IoU¯	42.87 (+0.58)	61.35 (+0.33)	
DIoU-de	W3-FM	IoU¯	42.53 (+0.24)	61.41 (+0.39)	
DIoU-de	A3-FM	IoU¯	42.74 (+0.45)	61.07 (+0.05)	
WIoU-de	W3-FM	IoU¯	42.63 (+0.34)	61.22 (+0.20)	
WIoU-de	A3-FM	IoU¯	42.90 (+0.61)	61.45 (+0.43)	

Similar to AIoU-v2-conf loss, we conducted a simple comparison experiment. In Table 7, the introduction of A3-FM(conf) in IoU, DIoU-de, and WIoU-de losses not only steadily improves the mAP value but also shows a stronger ability to enhance AP50 compared to A3-FM(IoU).

Table 7 Performance loss of each IoU loss under the A3-FM (IoU) and A3-FM (conf) on the MS-COCO (the best results are shown in bold).

BBR loss	FMs	Baseline	mAP	AP50	
IoU	None	None	42.29	61.02	
IoU	A3-FM	IoU¯	42.87 (+0.58)	61.35 (+0.33)	
IoU	A3-FM	conf¯	42.72 (+0.43)	61.61 (+0.59)	
DIoU-de	A3-FM	IoU¯	42.74 (+0.45)	61.07 (+0.05)	
DIoU-de	A3-FM	conf¯	42.72 (+0.43)	61.26 (+0.24)	
WIoU-de	A3-FM	IoU¯	42.90 (+0.61)	61.45 (+0.43)	
WIoU-de	A3-FM	conf¯	42.73 (+0.44)	61.49 (+0.47)	

Finally, we used the FM of the AIoU-v4 loss to reweight the BBR loss. As shown in Table 8, using both IoU and confidence as the weighting curve for BBR losses further enhances the mAP value, validating the role of A3-FM(conf). Additionally, when we monitored the IoU and confidence values during training, we observed that the A4-FM reweighting of IoU loss led to higher mAP values, even though the overall IoU values remained low, as shown in Fig. 5. We speculate that this phenomenon is due to AIoU-v4 loss effectively masking labeling errors during the training process.

Table 8 Performance loss of each IoU-based under AioUv4 focusing mechanisms on the MS-COCO (the best results are shown in bold).

BBR loss	FMs	mAP	AP50	
IoU	None	42.29	61.02	
IoU	A4-FM	42.98 (+0.69)	61.63 (+0.61)	
DIoU-de	A4-FM	42.93 (+0.64)	61.56 (+0.54)	
WIoU-de	A4-FM	43.08 (+0.79)	61.44 (+0.42)	

Comparison study

In “Ablation Study”, we analyzed in detail the impact of BBR loss on model regression under various modification methods. Here, we conduct a comparative experiment to evaluate several BBR losses mentioned earlier against our proposed AIoU losses. As shown in Table 9, our method achieves the most significant improvement. Compared to WIoU-v3, which had the best improvement effect among the previously proposed methods, AIoU-v4 achieves more than twice the improvement. As shown in Fig. 6, on the VOC2012 dataset, AIoU-v3 and AIoU-v4 significantly improve the mAP values of the model for medium-quality categories. Specifically, AIoU-v4 increases the mAP by 3.43% for the aerial category and by 3.33% for the bottle category, among others. Additionally, AIoU-v4-NMS, which utilizes our designed post-processing algorithm, further enhances the performance of BBR.

Table 9 Performance of each bounding box loss on the MS-COCO and VOC2012 (as shown in bold, AIoU-v4 loss achieved the best performance among all BBR losses).

BBR loss	MS-COCO	VOC2012	
	mAP	AP50	AP75	mAP	AP50	AP75	
IoU	42.29	61.02	45.03	60.34	80.85	65.09	
GIoU	42.55 (+0.26)	60.91 (−0.11)	45.37 (+0.34)	60.66 (+0.32)	80.76 (−0.09)	65.20 (+0.11)	
DIoU	42.49 (+0.20)	60.92 (−0.10)	45.51 (+0.48)	60.41 (+0.07)	80.54 (−0.31)	65.09 (+0.00)	
CIoU	42.44 (+0.15)	61.20 (+0.18)	45.20 (+0.17)	60.38 (+0.04)	80.91 (+0.06)	65.25 (+0.16)	
EIoU	42.26 (−0.03)	61.12 (+0.10)	49.66 (+4.63)	60.46 (+0.12)	80.37 (−0.48)	65.34 (+0.25)	
SIoU	42.42 (+0.16)	61.09 (+0.07)	45.16 (+0.13)	60.65 (+0.31)	80.90 (+0.05)	65.14 (+0.06)	
WIoU-v1	42.37 (+0.08)	61.18 (+0.16)	45.12 (+0.09)	60.37 (+0.03)	80.81 (−0.04)	65.03 (−0.06)	
WIoU-v2	42.45 (+0.16)	60.95 (−0.07)	45.28 (+0.25)	60.68 (+0.33)	80.82 (−0.03)	65.18 (+0.09)	
WIoU-v3	42.50 (+0.21)	61.16 (+0.14)	45.41 (+0.38)	60.72 (+0.38)	80.50 (−0.35)	65.44 (+0.35)	
AIoU-v1	42.44 (+0.15)	62.70 (+1.68)	47.07 (+2.07)	60.62 (+0.28)	80.88 (+0.03)	65.58 (+0.49)	
AIoU-v2	42.69 (+0.40)	61.13 (+0.11)	45.55 (+0.52)	60.70 (+0.36)	80.81 (−0.04)	65.19 (+0.10)	
AIoU-v3	42.90 (+0.61)	61.45 (+0.43)	47.11 (+2.08)	61.08 (+0.74)	81.05 (+0.30)	65.54 (+0.45)	
AIoU-v4	43.08 (+0.79)	61.44 (+0.42)	46.11 (+1.08)	61.35 (+1.01)	81.09 (+0.34)	66.20 (+1.11)	
AIoU-v4-NMS ( θ2=0.78)	43.15 (+0.86)	61.48 (+0.46)	46.19 (+1.16)	61.46 (+1.12)	81.15 (+0.40)	66.41 (+1.32)	

Figure 6 mAP for each class on the VOC2012.

Conclusion

In this article, we carefully analyzed the impact of BBR loss design on simulated bounding box regression and the YOLOv7 detector, describing their advantages and weaknesses. Our findings and improvements are as follows: (1) We found that the separate computation of the central regression terms x and y in the BBR loss facilitates bounding box regression. However, none of the previous BBR losses have been optimized for this issue. (2) Suppressing the gradient of low-quality samples is more reasonable than suppressing the gradient of high-quality samples, as it is more conducive to addressing dataset imbalances. (3) We proposed an adaptive dynamic non-monotonic focusing mechanism, which addresses the limitations of previous FMs that are not easily transferable and unsuitable for different regression stages and sample quality benchmarks. We also created a weighted curve that distinguishes categories, which other methods have not considered. (4) Existing works only use IoU to judge high-quality, medium-quality, and low-quality samples and do not utilize more useful data to improve the model’s performance in bounding box regression. We incorporated confidence to co-weight the BBR losses, effectively achieving hard sample mining and improving performance. (5) Finally, we proposed a post-processing algorithm that combines bounding box fusion and non-maximum suppression to further enhance the localization performance of object detection algorithms.

Although the algorithm proposed in this study has advantages, it has shown good performance in improving the recognition of medium-quality categories but has not significantly enhanced the recognition performance of low-quality categories. In future work, incorporating the training progress as a reference factor for FM could be considered so that the model can focus more on low-quality samples in the early stages.

Additional Information and Declarations

Competing Interests

Author Contributions

Data Availability

The authors declare that they have no competing interests.

Yanchen Zhu performed the experiments, analyzed the data, performed the computation work, authored or reviewed drafts of the article, and approved the final draft.

Wenhua Zheng conceived and designed the experiments, performed the experiments, analyzed the data, performed the computation work, prepared figures and/or tables, authored or reviewed drafts of the article, and approved the final draft.

Jianqiang Du analyzed the data, authored or reviewed drafts of the article, and approved the final draft.

Qiang Huang analyzed the data, prepared figures and/or tables, authored or reviewed drafts of the article, and approved the final draft.

The following information was supplied regarding data availability:

The data is available at Zenodo: Wenhua-Zheng. (2024). Wenhua-Zheng/AIOU: SourceData.v2.0 (v.2.0). Zenodo. https://doi.org/10.5281/zenodo.13132177.

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
