# Peer review of "Autonomous intersection over union (IoU) loss: adaptive dynamic non-monotonic focal IoU loss"

_PeerJ Computer Science, doi:10.7717/peerj-cs.2347_

## Round 0.1 · original submission · Major Revisions

Dear Authors
Based on the reviewers' reports, your paper requires significant revisions before it can be considered for publication in PEERJ Computer Science.

The major issues are listed in what follows:

1) It is necessary to explain how the main innovative component of this paper, the AIoU loss, is effectively integrated into the object detection processes described in Sections 1.2, 1.3, and 1.4 and why it performs better than other methods.
2) The experimental results could be clearer and more sufficient to show the proposed method's main advantages and contributions. Some additional experiments need to be conducted to strengthen the conclusion.

Furthermore:
1) The sentences and grammar of the manuscript should be carefully checked and revised.
2) Some figures need to be clearer and need to be modified. Tables should be normalized.

Reviewer 1 ·

Basic reporting

This paper analyzed the original BBR loss and proposed AIoU-v1 loss, and designed AIoU-v3 with adaptive dynamic non-monotonic FM.
1. The sentences and grammar of the manuscript should be carefully checked and revised.
2. Some figures are not clear enough and need to be modified. Tables should be normalized.
3. You should be more standardized and reasonable to summarize and review the relevant work. Some related works should be discussed in this paper: --Active learning for deep tracking, --AIoU: Adaptive bounding box regression for accurate oriented object detection, Self-supervised Deep Correlation Tracking

Experimental design

4. The experimental results are very unclear and not enough to show the main advantages and contributions of the proposed method.
5. Experiment validations are not convincing. Some additional experiments need to be conducted to make its conclusion stronger.

Validity of the findings

no comment

Additional comments

6. Please consider discussing and analyzing the limitations of the proposed method.

Reviewer 2 ·

Basic reporting

The content of the article is mostly understandable. A sufficient background is provided on the loss function, which is the primary topic of the paper. Figures and tables are professionally structured. However, the captions of all figures are very short and do not describe the content of the figures. There are enough experiments and results.

Experimental design

The authors have described the limitations of existing loss functions for bounding box regression and compared them mathematically and numerically. However, following different versions of loss functions was difficult, e.g., v1, v2, etc. Therefore, I recommend putting together a table that mentions a list of loss functions, their formula, and their different versions.

1. I could not find the definition of FM.
2. Which loss function is defined by Eq 3: GIOU or DIOU?
3. Which model is used for regression for the simulation experiments in Sec. 3.1?
4. What do you mean by final error?
5. What does the suffix "-de" signify? Does it indicate the decoupling of height and width from the denominator of the loss function?
6. Does the IOU threshold for filtering the samples (low and high-quality) depend on the dataset used for training the model?
7. Highlight the best model in bold color in each result table.

Validity of the findings

The authors conclude their findings and compare with state-of-the-art methods.

Reviewer 3 ·

Basic reporting

see attached file

Experimental design

see attached file

Validity of the findings

see attached file

Additional comments

see attached file

Annotated reviews are not available for download in order to protect the identity of reviewers who chose to remain anonymous.

---

## Round 0.2 · accepted · Accept

Dear Authors,

Your paper has been accepted for publication in PEERJ Computer Science. The comments of the reviewers who evaluated your manuscript are included in this letter. I ask that you make minor changes to your manuscript based on those comments, before uploading final files. Thank you for your fine contribution.

Reviewer 1 ·

Basic reporting

no comment'

Experimental design

no comment'

Validity of the findings

no comment'